# Caspase Inhibition as a Possible Therapeutic Strategy for Pemphigus Vulgaris: A Systematic Review of Current Evidence

**DOI:** 10.3390/biology11020314

**Published:** 2022-02-16

**Authors:** Sanna Huda, Bethany Chau, Chuanqi Chen, Herman Somal, Neiloy Chowdhury, Nicola Cirillo

**Affiliations:** Melbourne Dental School, The University of Melbourne, Carlton, VIC 3053, Australia; snhuda@student.unimelb.edu.au (S.H.); bchau@student.unimelb.edu.au (B.C.); chuanqic@student.unimelb.edu.au (C.C.); hsomal@student.unimelb.edu.au (H.S.); neiloyc@student.unimelb.edu.au (N.C.)

**Keywords:** pemphigus vulgaris, acantholysis, apoptosis, caspase, caspase inhibitor

## Abstract

**Simple Summary:**

Pemphigus vulgaris is a potentially fatal disease characterised by blister formation affecting the skin and mouth. The mechanisms of blister formation may involve a biological process called apoptosis—a type of cell death—and some death-associated molecules known as caspases. Our review of the existing literature shows that caspase inhibitors exhibit an inhibitory effect on PV-induced apoptosis formation in vitro. In particular, activity of caspase 1 and caspase 3 is essential for the development of PV in vitro and in vivo. However, a majority of in vivo studies assessing caspase inhibition in PV models have a high risk of bias.

**Abstract:**

Background: Pemphigus vulgaris (PV) is an IgG-mediated autoimmune disease characterised by epithelial cell–cell detachment (acantholysis) resulting in mucocutaneous blistering. The exact pathogenesis of blister formation is unknown and this has hampered the development of non-steroidal, mechanism-based treatments for this autoimmune disease. This systematic review aims to investigate the role of caspases in the pathogenesis of PV to inform the choice of more targeted therapeutic agents. Methods: A systematic search of MEDLINE/PubMed and Scopus databases was conducted to identify eligible studies. Multiple phases of inclusion and exclusion of the primary articles were conducted in pairs, and studies were recorded and analysed according to the latest version of the preferred reporting items for systematic reviews and meta-analyses (PRISMA). Risk of bias assessment was conducted for extracted in vivo animal intervention studies using SYRCLE’s risk of bias tool. Results: Eight articles from a total of 2338 in vitro, in vivo, and human studies met the inclusion criteria, with a high degree of inter-rater reliability. By and large, the results show that caspase activation was pathogenic in experimental PV because pan-caspase inhibitors could block or reduce PV acantholysis and blistering in vitro and in vivo, respectively. The pathogenic pathways identified involved caspase-1 and caspase-3. One study failed to show any improvement in the PV model with a caspase inhibitor. The majority of animal studies had high or unclear risk of bias. Conclusion: There are consistent data pointing towards a pathogenic role of caspase activation in PV acantholysis. However, high-quality evidence to confirm that caspase inhibition can prevent PV-induced blistering in vivo is limited. Therefore, further research is required to test the preclinical efficacy of caspase inhibitors in PV.

## 1. Introduction

Pemphigus Vulgaris (PV) is a life-threatening IgG-mediated autoimmune disease characterised by mucocutaneous blistering resulting from detachment of keratinocytes, known as acantholysis. This affects 0.1–0.5 per 100,000 people globally and mortality rate in these patients can reach 60–90% without effective disease management [1]. The use of corticosteroids significantly reduces the mortality rate to 10% [1]. However, the long-term use of corticosteroids leads to severe adverse effects, including peptic ulcer disease, adrenal insufficiency, osteoporosis, and can increase treatment-related morbidity [1,2]. Even with these adverse effects, corticosteroids, with or without immune suppressants, remain the first-line therapy due to their rapid effect in treating PV [3]. Therefore, the search for novel, more specific therapeutic targets that improve PV treatment efficacy and minimise adverse effects is necessary; this cannot be achieved without gaining a deeper understanding of how acantholysis and blister formation occurs in PV.

The pathophysiological mechanisms of PV are still incompletely understood, though the etiopathogenesis is thought to be multifactorial. This involves a complex series of events elicited by binding of PV-IgG to their target antigens, as well as by non-IgG serum factors. One of the proposed pathomechanisms, apoptolysis [4], reconciles the findings that link apoptotic and acantholytic pathways that determine blister formation. Specifically, auto-antibody binding to a myriads of pemphigus autoantigens leads to initiation of apoptotic enzyme cascades in keratinocytes, resulting in basal cell shrinkage and supra-basal acantholysis [4]. Studies show that there is a notable increase in the expression of both caspases and apoptotic cells in PV patients [5,6,7,8,9], and that apoptosis in PV takes place via both intrinsic and extrinsic pathways. The extrinsic apoptotic pathway in PV includes the Fas receptor (FasR) which is located at the cell surface of keratinocytes [10]. Apoptosis is also induced through the intrinsic pathway via mitochondrial dysfunction and release of cytochrome c to activate caspase-9 and caspase-3, which in turn trigger apoptosis of the cell.

Although there is uncertainty on the relationship between apoptosis and acantholysis, it has been shown that the inhibition of caspases can prevent both processes in PV [11,12]. Given that both pathways lead to caspase activation and are known to be activated in PV [11,12], it would be reasonable to focus on caspase activation rather than apoptosis with an aim of developing mechanism-based pharmacological treatments. Furthermore, while caspases target a large spectrum of molecules, they also cleave desmosomal proteins including desmogleins [13]. Proteolytic processing of desmosomal cadherins leads to cell–cell detachment and cell shrinkage [13]. It is plausible, therefore, that PV-IgG or PV sera induce the pathological activation of caspases in keratinocytes which may lead to acantholysis [11,12]. To this regard, a recent scoping review discussing the possible pathogenic mechanisms of PV noted the caspase pathway as a possible key driver of blister formation [14].

The vital function of caspases in the intracellular cascade of acantholysis in PV makes this class of molecules a potential pharmacological target for novel mechanism-based treatments of the disease. Therefore, the aim of this systematic review was to assess the evidence pertaining the role of caspases and their inhibition in preventing acantholysis and blistering in PV.

## 2. Methods

### 2.1. Search Strategy

This systematic review reports results according to the methodology outlined by the 2020 version of the Preferred Reporting Items for Systematic Reviews and Meta-analyses (PRISMA) guidelines [15]. This review was designed to analyse broad evidence collected in all experimental settings (in vitro, in vivo, and human studies), and hence was not eligible for registration with PROSPERO.

The following search strategy was applied to Scopus and PubMed databases to collate articles discussing PV and caspases and associated terms. This search was conducted in May 2021 using the following search string:

(Pemphigus vulgaris) AND (Caspas* OR CASP OR apopto* OR metalloproteinas* OR metalloproteas* OR MMP).

### 2.2. Eligibility Criteria

As this review aims to determine the role of caspases in PV, articles were included if caspase inhibition or blocking demonstrated an effect (or lack thereof) on PV models. Thus, the criteria for inclusion were satisfied if a study involved pharmacological inhibition, knock-out, knock-down, silencing or inactivation of caspases in a PV model or in PV patients. The pathogenicity criterion was satisfied whereby caspase inhibition resulted in the absence of or a reduced PV phenotype in vivo, acantholysis/intercellular detachment in vitro, or clinical manifestations in PV patients. Studies were excluded if they were non-English, review articles, reported on forms of pemphigus other than PV, did not report on caspases or did not report on the pathogenicity criterion. There was no time restriction.

### 2.3. Data Selection and Collection

During the initial search, non-English articles and reviews were filtered out and excluded. Search results for each database were then downloaded into separate Excel spreadsheets. A total of 4 independent reviewers title-screened the articles. A pilot kappa score for title screening of the first 20 articles for each database was generated by a blinded 5th reviewer prior to formal screening to ensure a high degree of inter-examiner reliability. Cohen’s Kappa coefficient was calculated to be 1 (Scopus) and 0.875 (PubMed), demonstrating a perfect and strong level of agreement respectively.

Subsequently, for each database (Scopus and PubMed), a complete title screen was conducted by 2 independent reviewers to determine if an article was relevant. Any disagreements and confusion on articles that could not be confidently excluded were resolved by the 5th reviewer and the research supervisor. Title screened articles from both databases were then collated into a single Excel spreadsheet and duplicates were removed. A combined title and abstract screening was then conducted. Total articles were allocated into halves, and the articles in each half were assessed by 2 independent reviewers. Articles were removed according to the same criteria used in the title screening. Kappa scores were then generated for the combined title and abstract screening of each half to determine inter-examiner reliability. Cohen’s Kappa coefficient was calculated to be 0.825 and 0.77, demonstrating a strong and moderate level of agreement respectively.

Full-text screening for each article was performed by 2 independent reviewers (completed amongst 5 reviewers). Articles were selected according to the inclusion criteria. Data was extracted from the selected articles and tabulated.

### 2.4. Quality Assessment

Risk of bias assessment was conducted for extracted in vivo animal intervention studies using SYRCLE’s risk of bias tool [16]. This was performed to determine the methodological quality of animal experiments. SYRCLE’s risk of bias tool contains 10 entries related to selection bias, performance bias, detection bias, attrition bias, reporting bias and other biases. Signalling questions are used for each entry and responses can be “yes” (low risk of bias), “no” (high risk of bias) and “unclear” (unclear risk of bias).

“Unclear” was used when the information was not mentioned or could not be deciphered from the article. Two independent reviewers completed the risk of bias assessment for each in vivo article and any confusions or disagreements were resolved by another independent reviewer.

## 3. Results

### 3.1. Search Results

There were 2173 and 165 records identified in Scopus and PubMed, respectively. Out of these studies, eight met the inclusion criteria and were included in the qualitative synthesis (Figure 1). The strategies for caspase inhibition used in these papers were categorised into genetic and pharmacologic, the latter including pan-caspase inhibitors (*n* = 5), caspase-1 inhibitors (*n* = 1) and caspase-3 inhibitors (*n* = 1). Only one study [17] used genetic knock-out mice strain to inhibit a caspase, with the remaining using a chemical agent [6,11,12,18,19,20,21]. More than half of these studies (*n* = 5) used in vivo PV models only and the remaining (*n* = 2) using in vitro PV models only. There was one study [18] that utilised both an in vitro and in vivo PV model. All studies used either mice models or human keratinocyte cell lines that were injected/incubated with human PV sera, PV-IgG, or anti-Dsg3 IgG (such as AK23 antibody).

### 3.2. Pan-Caspase Inhibition

There were five studies that utilised pan-caspase (broad spectrum caspase) inhibitors (Table 1). Out of five studies, three showed a statistically significant (*p* < 0.05) inhibitory effect on PV [6,11,18]. Four studies showed a decrease in acantholysis, apoptotic cells, apoptotic biomarkers, and/or lesions [6,11,18,19]. All four in vivo studies performed a skin biopsy on mice injected with human PV IgG [6,11,18,19]. There were two in vitro studies that used a human cell line incubated with human PV sera [18,21].

There was 1 study that showed no improvement in the PV model with a caspase inhibitor. Schmidt et al. incubated a pan-caspase inhibitor in PV-induced HaCaT and NHEK cells [21]. Upon immunostaining investigation of the cells, the inhibitor showed no difference in cell dissociation and DSG-3 fragmentation when compared to the PV-induced HaCaT and NHEK cells with no inhibitor.

### 3.3. Caspase-1 Inhibition

Only one in vitro study specifically used a caspase-1 inhibitor (Table 2) [12]. Wang et al. [12] showed that the caspase-1 inhibitor prevents apoptosis and lesion formation in HaCaT cells incubated with human PV IgG.

### 3.4. Caspase-3 Inhibition

There were two studies from the search that employed caspase-3 inhibition, both of which were in vivo (Table 3) [17,20]. Hariton et al. [17] utilised a caspase-3 genetic knock-out in a mice model and showed approximately a 50% decrease in hair follicle PV blisters and a decrease in caspase biomarkers in the caspase-3 knockout mice compared to the control mice. Using the pharmacological caspase-3 inhibitor, Luyet et al. [20] showed an approximate 15% decrease in hair follicle PV blisters.

### 3.5. Risk of Bias Assessment

A risk of bias questionnaire was undertaken for all in vivo studies that met the inclusion criteria and the results of the risk of bias assessment are listed in (Table 4). Most studies revealed unclear or high risks of bias. In particular, none of the studies described randomization of the animals/cages, sequence generation, blinding, or allocation concealment.

## 4. Discussion

The results of the present systematic review show that caspase inhibition can decrease blistering, apoptosis and/or acantholysis in mouse and human keratinocyte models of PV. However, assessment of the quality of in vivo studies suggests that there is a high risk of bias for most animal experiments. Therefore, the overall evidence that caspases can effectively inhibit PV phenotype in vivo cannot be established with confidence.

### 4.1. Effect of Caspase Inhibition

The studies included targeting pan-caspases or a specific caspase (caspase-3, caspase-1) in an attempt to prevent PV acantholysis. Prevention of cell–cell detachment and/or blister formation was achieved for pan-caspase inhibitors in the form of: (1) absence or reduction of PV blistering [6,11,19]. (2) absence or reduction of acantholysis or cell–cell adhesion strength [17,18,19]. Additionally, Schmidt et al. [21] reported that pan-caspase inhibition had no effect on cell dissociation. Caspase-3 inhibition only reduces blisters, and failed to fully prevent PV phenotype [17,20]. Interestingly, Wang et al. [12] reported that caspase-1 inhibition can prevent cell death and lesion formation; however, other caspases may be responsible for this effect as the inhibitor used in the study (YVAD-CHO) is accepted to be non-specific. Additionally, this study did not report *p*-values and results on blockage of lesions in cell cultures, which undermines the reliability of their results [12]. Whilst it is interesting that desmosomal proteins can be cleaved by caspases during apoptosis [11,22], the type of proteolytic processing depends on the apoptotic stimulus [23]. Therefore, it is possible that PV IgG or sera, particularly when used at different concentrations, induce distinct caspase activation cascades that result in varying degrees of cell–cell dyscohesion.

Some studies were unable to demonstrate complete inhibition of PV with caspase inhibition [17,18,20,21]. These findings may indicate that other signalling molecules may be driving the disease process even in the presence of caspase inhibition. Therefore, the effect of caspase inhibition in clinically preventing disease adequately is questionable. This suggests that there is currently insufficient evidence to focus preclinical and clinical research interest into a specific caspase inhibitor as various molecular pathways are capable of inducing blistering and/or acantholysis.

Current literature on caspase inhibition in PV indicates that apoptotic pathways are involved. The Fas/Fas ligand system may be involved in keratinocyte apoptosis in PV through downstream activation of caspase-8 [24]. Additionally, executioner caspase inhibition can lead to significant reductions in PVIgG-dependent elevation of cytochrome c [25], and hence cell death. These studies, in part, support the concept of apoptolysis; however, more research is required to ascertain the occurrence of acantholysis.

### 4.2. Contradicting Findings in Current Literature

Contradicting TUNEL results were reported by Wang et al. [12] and Schmidt et al. [21], which may be explained by the differences in the methods and materials used. The use of TUNEL assay as an indicator for acantholysis relies on an assumption that there is a direct relationship between apoptosis, DNA fragmentation and acantholysis. Although both Wang et al. and Schmidt et al. followed the same Promega protocol for TUNEL assay, different results were reported. Wang et al. [12] observed DNA fragmentation in PV-IgG-induced acantholysis; whereas, Schmidt et al. [21] showed that no TUNEL reactivity was observed. Both studies used the PV patient skin lesion biopsies, but Wang et al. [12] did not include relevant information about tissue processing after the biopsy, while Schmidt et al. [21] recorded the process in detail. Tissue processing may explain the discrepancies with the results.

Use of other techniques may be more accurate for determining acantholysis without reliance on biomarkers of apoptosis. The use of cytokeratin retraction in the Schmidt et al. [21] article may be a better direct indicator of acantholysis. However, caspase inhibitor z-VAD-fmk could not prevent the PV-IgG mediated acantholysis; this contradicts the findings described in other studies. Additionally, a control group is necessary to confirm the efficacy of the z-VAD-fmk agent.

### 4.3. Limitations and Considerations for Future Studies

The dosages used in experimental models is an area for potential improvement as it can aid in predicting prevention of blistering and acantholysis. All included studies except Wang et al. [12] inoculated PV models with a single caspase inhibitor and pathogenic agent concentration. The use of multiple dosage concentrations allowed for a dose-response relationship to be established in the Wang et al. study [12]. As this was the only study that addressed varying dosage concentrations, a reliable dose response effect has yet to be established. The large inter-study variations with the type and concentration of PV-IgG and caspase inhibition also makes it difficult to translate this information to humans in addition to gauging a relationship between efficacy and dosage. Hence, future research may consider establishing a consistent dosing scheme to facilitate investigation into potential doses for human studies.

Current literature focuses on in vivo mouse models and in vitro human keratinocytes to provide further insight on caspase pathway involvement in PV. However, this presents a major challenge when attempting to translate these research findings to humans as expression of key molecules differs between humans and mice, which may lead to different mechanisms for cell cohesion and signalling. For example, expression of the desmoglein family of molecules differs between humans and mice (DSG1-4, Dsg1-6) [26], so the results obtained from some of the studies may not be reliable in extrapolation in humans. These differences may result in involvement of other mechanisms affecting cell cohesion and signalling. We suggest that a dog model might be more usable in investigating blistering dermatoses in further studies, as desmoglein expression in dogs is similar to humans. Similarly, results from in vitro studies may not completely correlate to the human body’s response to caspase inhibition. Human PV skin samples should be included in further research as it appreciates the complexity of human physiology and would provide data that is more practical for further research on potential relevant therapeutic agents. This would provide more relevant information as opposed to in vitro experiments on human keratinocytes and in vivo PV models. However, it is important to recognise that human samples may lead to inconsistent results due to patient-to-patient variability as PV pathophysiology is heterogeneous [15]. Therefore, standardization of human PV skin samples is required such that the research produced is consistent.

The studies assessed and other current literature indicate that it may be worthwhile to consider other molecules in future PV research. Arrendondo et al. discussed the potential induction of acantholysis by calpains via the oncotic pathway of programmed cell death [18]. Some studies have also mentioned that the upstream Fas apoptotic pathway may play a role in the acantholytic process of PV [7,27]. This suggests that future research could consider focusing investigation on other potential molecular targets that are also involved in cell death signals, such as c-myc, p38MAPK and EGFR inhibitors, which have demonstrated improvements in PV [28,29,30]. Therefore, it may be useful to investigate these molecules and their broader pathways to explore other potential targets for treatment. This is particularly relevant for PV research because caspase inhibitors are not currently used in clinical settings. In other chronic skin diseases in which caspases play an important role, such as psoriasis, caspase inhibitors have been used for in Phase I and II clinical trials (clinical trial: NCT00205465) but have not been approved for clinical use to date [31]. An alternative approach could be to target molecules that are relevant to PV pathophysiology and that also share the same caspase-mediated cell death pathways, such as FasL and MMPs. The administration of anti-FasL antibodies after PVIgG injection blocks blister formation in wild-type mice, and mice lacking secreted soluble FasL develop a milder disease upon PVIgG injection [32]. Similarly, we have shown that MMP-9 is overexpressed in both in vivo and in vitro models of PV [33] and that MMP inhibition is a promising route to non-immunosuppressive PV treatment [34].

In our review, a risk of bias assessment was undertaken for all in vivo studies that met the inclusion criteria (Table 4). The purpose of assessing the risk in these studies was to evaluate the quality of each study to determine whether the results can be generally accepted by the scientific community. The risk of bias assessment indicated that all evaluated experiments were not blinded to the investigators. This can generate a significant amount of bias when the investigators are choosing the tissue sections; without blinding, researchers are capable of subjectively choosing the ‘preferable’ outcome when they are aware of which experimental group they are observing. Moreover, many of the studies did not mention the experimental conditions that form part of SYRCLE’s assessment criteria; this resulted in several “unclear” responses leading to a limited bias assessment. Additionally, one study [19] did not explicitly provide data for the outcomes of interest which undermines the reliability of the results.

## 5. Conclusions

This systematic review suggests that caspases are pathogenically involved in PV, but it is still unclear whether they are solely responsible for driving acantholysis. Although there is a sizable body of evidence available reporting on the potentially curative effect of caspase inhibition in PV, the studies assessed in this review reported varying results on the effect of caspase inhibition on acantholysis, apoptosis and blistering lesions. These studies have also demonstrated the need for more rigorous and controlled methodological measures to limit bias. Given that current literature regarding the role of caspases in PV demonstrates that caspase inhibition may reduce acantholysis in PV, further well-designed studies are required to confirm this relationship. Additionally, there are no caspase inhibitor medications clinically approved for use [35]. Thus, if caspase inhibitors are to be considered in future therapeutic treatment of PV, safety concerns and side effects in humans would need to be thoroughly evaluated.

## Figures and Tables

**Figure 1 biology-11-00314-f001:**
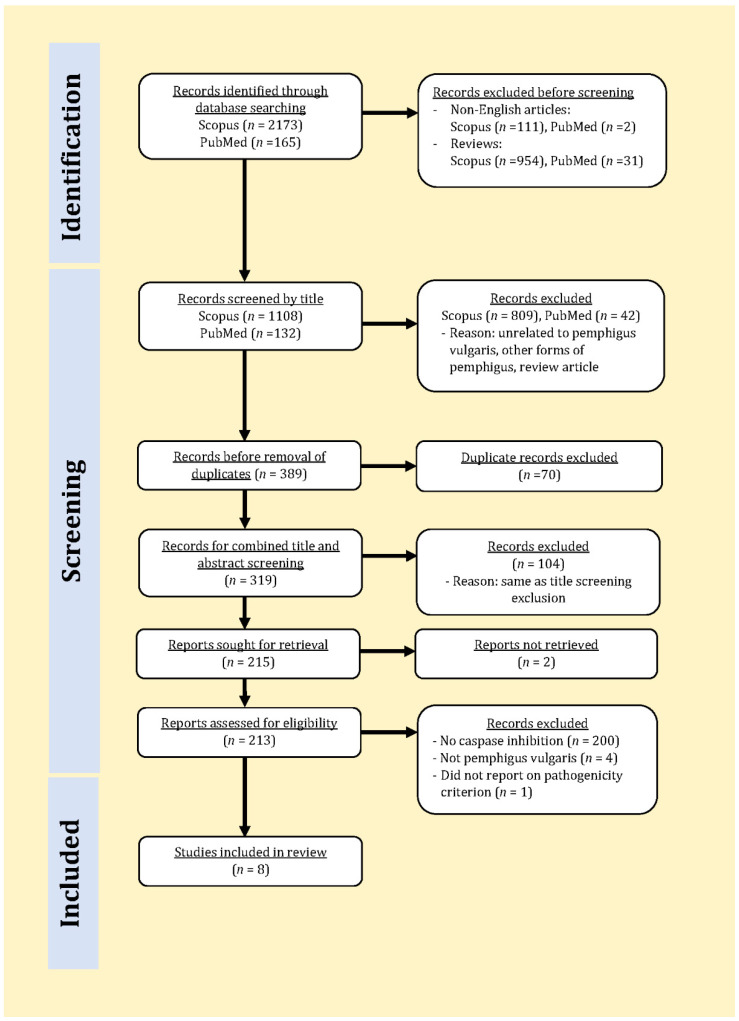
Flowchart of data selection process in accordance with 2020 PRISMA guidelines. Number of records assessed (**left side**) and excluded (**right side**) are reported.

**Table 1 biology-11-00314-t001:** Studies using pan-caspase inhibitors.

Author, Year	Type of Study	PV Model with Concentration of PV Sera/PV Antibody	Details of Inhibition (Type and Concentration, if Relevant)	Outcome
Arredondo et al. [18]	IN VITRO, IN VIVO	IN VITRO: Human PV IgG from 2 patients (PVIgG1b or PVIgG-2b) (1 mg/mL) incubated in human keratinocyte monolayers IN VIVO: Mice injected (intraperitoneal) with human PV IgG from two different patients (PVIgG-1b or PVIgG-2b) (0.1 mg/g body weight)	IN VITRO: Caspase inhibitors DEVD-CHO (10 μM), Z-DEVD-FMK (10 μM), and Z-DCB-MK (100 μM), given alone or as a mixture. IN VIVO: 25 μg/g body weight MDL28170 and 10 μg/g body weight Z-DCB-MK	IN VITRO: Caspase inhibitors given alone or as a mixture, can completely inhibit acantholysis in keratinocyte monolayers treated with PVIgG-2b (*p* < 0.05), and caused moderate decrease of acantholysis of PVIgG-1b (*p* > 0.05). IN VIVO: Z-DCB-MK significantly (*p* < 0.05) reduced extent of acantholysis induced by PVIgG-2b by approximately 50%.
Schmidt et al. [21]	IN VITRO	HaCaT and NHEK cell lines incubated with human PV IgG (0.15 mg/mL)	Pan-caspase inhibitor z-VAD-fmk (20 mM)	Immunostaining analysis of the HaCaT and NHEK cells incubated with PV IgG showed no improvement in cell dissociation and DSG-3 fragmentation.
Pacheco-Tovar et al. [6]	IN VIVO	Mice injected (intraperitoneal) with human PV IgG (1 mg/g body weight)	Pan-caspase inhibitor: AC- DEVD-CMK (20 mM)	Caspase inhibitor prevents macroscopical and histological blistering, apoptosis and acantholysis (*p* < 0.0037).
Pretel et al. [11]	IN VIVO	Mice injected with human PV IgG (2 mg/g bodyweight)	Pan-caspase inhibitor: cpmVAD-CHO (1.6 µg/g bodyweight)	Caspase inhibitor showed complete absence of PV lesions from histological and clinical examination.
Gil et al. [19]	IN VIVO	Mice injected with human PV IgG (2 mg/g bodyweight)	Pan-caspase inhibitor (cpmVAD-CHO) (1.6 µg/g body weight)	Caspase inhibitor led to absence of clinical PV lesions and suprabasal acantholysis

Abbreviations: PV (pemphigus vulgaris), NHEK (normal human epidermal keratinocytes).

**Table 2 biology-11-00314-t002:** Studies using caspase-1 inhibitors.

Author, Year	Type of Study	PV Model with Concentration of PV Sera/PV Antibody	Details of Inhibition (Type and Concentration, if Relevant)	Outcome
Wang et al. [12]	IN VITRO	HaCaT cell line and skin organ cultures incubated with human PV IgG (2.5 mg/mL)	YVAD-CHO (0–20, 100 nM)	Caspase inhibitor prevents PVIgG-induced cell death and lesion formation via histological analysis.

Abbreviations: PV (pemphigus vulgaris).

**Table 3 biology-11-00314-t003:** Studies using caspase-3 inhibitors.

Author, Year	Type of Study	PV Model with Concentration of PV Sera/PV Antibody	Details of Inhibition (with Concentrations, if Relevant)	Outcome
Hariton et al. [17]	IN VIVO	Mice injected with AK23 antibody (12.5 μg/g body weight)	Caspase-3 genetic knock-out (KO) (-/-)	The caspase-3 KO showed approximately a 50% reduction in hair follicle PV blisters (*p* < 0.05) and a significant decrease in pro-caspase-3 biomarkers (*p* < 0.05) when compared to PV wildtype and heterozygotes.
Luyet et al. [20]	IN VIVO	Mice injected with AK23 antibody (12.5 μg/g body weight)	Caspase-3 inhibitor Ac-DEVD-CMK (6 μg/g)	The caspase-3 inhibitor showed approximately a 15% decrease in hair follicle PV blisters.

**Table 4 biology-11-00314-t004:** SYRCLE’s risk of bias tool used to assess included in vivo animal intervention studies (green, low risk of bias; orange, high risk of bias; yellow: unclear risk of bias).

Arredondo et al. [16]	Pacheco-Tovar et al. [6]	Hariton, W.V. [15]	Pretel et al. [9]	Gil et al. [17]	Luyet et al. [18]	Signalling Question	Type of Bias and Domain
Unclear	Unclear	Unclear	Unclear	Unclear	Unclear	Did the investigators describe a random component in the sequence generation process?	Selection bias (sequence generation)
Yes	Yes	Yes	Yes	Yes	Yes	Was the distribution of relevant baseline characteristics balanced for the intervention and control groups?	Selection bias (baseline characteristics)
Unclear	Unclear	Unclear	Yes	Unclear	Unclear	Was the timing of disease induction adequate?	Selection bias (baseline characteristics)
Unclear	Unclear	Unclear	Unclear	Unclear	Unclear	Could the investigator allocating the animals to intervention or control group not foresee assignment due to one of the following or equivalent methods?	Selection bias (allocation concealment)
Unclear	Unclear	Unclear	Unclear	Unclear	Unclear	Did the authors randomly place the cages or animals within the animal room/facility?	Performance bias (random housing)
Yes	Yes	Yes	Yes	Yes	Yes	Is it unlikely that the outcome or the outcome measurement was influenced by not randomly housing the animals?	Performance bias (random housing)
Unclear	Unclear	Unclear	Unclear	Unclear	Unclear	Was blinding of caregivers and investigators ensured, and was it unlikely that their blinding could have been broken?	Performance bias (blinding)
Unclear	Unclear	Unclear	Unclear	Unclear	Unclear	Did the investigators randomly pick an animal during outcome assessment, or did they use a random component in the sequence generation for outcome assessment?	Detection bias (random outcome assessment)
Unclear	No	Unclear	Unclear	Unclear	No	Was blinding of the outcome assessor ensured, and was it unlikely that blinding could have been broken?	Detection bias (blinding)
No	No	No	No	Yes	Yes	Was the outcome assessor not blinded, but do review authors judge that the outcome is not likely to be influenced by lack of blinding?	Detection bias (blinding)
Unclear	Yes	Yes	No	Yes	Yes	Were all animals included in the analysis?	Attrition bias (incomplete outcome)
No	No	No	No	Yes	Unclear	Was the study protocol available and were all of the study’s pre-specified primary and secondary outcomes reported in the current manuscript?	Reporting bias (selective outcome reporting)
Yes	Yes	Yes	Yes	No	Yes	Was the study protocol not available, but was it clear that the published report included all expected outcomes (i.e., comparing methods and results)?	Reporting bias (selective outcome reporting)
Yes	Yes	Unclear	Yes	Unclear	No	Was the study free of contamination (pooling drugs)?	Other sources of bias
Yes	Yes	Yes	Yes	Yes	Yes	Was the study free of inappropriate influence of funders?	Other sources of bias
Unclear	Unclear	Unclear	Unclear	Unclear	Yes	Was the study free of unit of analysis errors?	Other sources of bias
Unclear	Unclear	Unclear	Unclear	Unclear	Unclear	Were design-specific risks of bias absent?	Other sources of bias
Unclear	No	No	Unclear	Unclear	No	Were new animals added to the control and experimental groups to replace drop-outs from the original population?	Other sources of bias

## Data Availability

Supplementary results are available with the article. All data items not included in the manuscript are available upon reasonable request to the corresponding author.

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
