# Peer review of "Caspase Inhibition as a Possible Therapeutic Strategy for Pemphigus Vulgaris: A Systematic Review of Current Evidence"

_biology, 2022, doi:10.3390/biology11020314_

Round 1

Reviewer 1 Report

In this systematic review authors gave an overwiev of the literature regarding the role of caspases in PV and finaly 8 papers were included. The paper is clearly and well written, the study was properly designed and all the limitations are indicated. Therefore I recomed this paper for publication. Before publication following should be corrected:

Line 43: "....rate to 10%.1 However,......"        Number 1 means what?

Line 140: It is better not to start the sentence with the number

Author Response

Thank you for your comments - all criticisms have been addressed. 

Reviewer 2 Report

This is a review paper on the current evidence on the role of caspase inhibitors in the management of pemphigus vulgaris. The title is informative, and the key words match the topic. The paper is objective and logically arranged, supported with a matching literature. The level of English is excellent. The methodology is presented clearly and illustrated with tables and a flowchart. The authors skillfully addressed the strengths and limitations of the review.

Major issues:
1. My advice would be to adhere to the HUGO Gene Nomenclature Committee guidelines. Anti-Dsg3, a style the authors use uniformly through the paper, is a mice protein, whereas anti-DSG3 is a human protein.
Please check the details here:
https://academic.oup.com/molehr/pages/Gene_And_Protein_Nomenclature
https://www.biosciencewriters.com/Guidelines-for-Formatting-Gene-and-Protein-Names.aspx

Minor issues:
1. It would be good to add "caspase inhibitor" as a 5th keyword and it is an official MeSH term.
2. All the tables should be provided with a legend explaining the abbreviations used.
3. In my opinion, there is exceedingly much data presented in the results. Please consider moving Table 4 to supplementary materials.
4. What is worth mentioning is that many of the studies were performed on mice models. Expression of the desmoglein family of molecules differs between humans and mice (DSG1-4, Dsg1-6), so the results obtained from some of the studies may not be reliable in extrapolation in humans. These differences may result in involvement of other mechanisms affecting cell cohesion and signaling. Dog model might be more usable in investigating blistering dermatoses in further studies, as the desmoglein expression in dogs is similar to humans. https://pubmed.ncbi.nlm.nih.gov/12787123/ 
5. Ref. 4. uses both the full name of the journal and abbreviation. The former is unnecessary.

Author Response

(The authors gave the same response as above.)

Reviewer 3 Report

Huda et al. summarize and compare 8 articles that investigated the use of pan-caspase inhibitors, caspase 3 inhibitors, or caspase 1 inhibitors on PV-IgG induced acantholysis and blistering. The studies were selected after strict literature search and other criteria that I am not familiar with and which I cannot evaluate. Studies were either in vitro (cell culture) or in vivo (mice). Huda et al. concluded that in 7 out of 8 studies, caspase inhibition showed some degree of benefit, however, studies were different in terms of compound and dosing etc. Furthermore, a certain degree of bias was detected.

The authors concluded that further research is required to test the preclinical and clinical efficacy of caspase inhibitors in PV.

Comments:

The summary of the 8 papers gives a nice overview on what was published in terms of PV and caspase inhibition, however, their general conclusion that more research is needed is a bit weak. The authors should include some more critical discussion on the clinical use of caspase inhibitors. Is any of the mentioned inhibitors approved for clinical use in humans? If not, are there other caspase inhibitors available for clinical use? For which pathologies are they used? What are the challenges/side effects etc. of caspase inhibition in humans? What is the authors opinion on the future use of caspase inhibitors in Pemphigus Vulgaris?

Author Response

(The authors gave the same response as above.)

Round 2

Reviewer 3 Report

My questions were answered.